# Influences of Crystallinity and Crosslinking Density on the Shape Recovery Force in Poly(ε-Caprolactone)-Based Shape-Memory Polymer Blends

**DOI:** 10.3390/polym14214740

**Published:** 2022-11-04

**Authors:** Ailifeire Fulati, Koichiro Uto, Mitsuhiro Ebara

**Affiliations:** 1Research Center for Functional Materials, National Institute for Materials Science, Tsukuba 3050044, Japan; 2Graduate School of Science and Technology, University of Tsukuba, Tsukuba 3058577, Japan; 3Graduate School of Advanced Engineering, Tokyo University of Science, Tokyo 1258585, Japan

**Keywords:** shape-memory polymer, crystallinity, crosslinking density, shape recovery force, energy storage capacity, polymer blends, semicrystalline polymer

## Abstract

Shape-memory polymers (SMPs) show great potential in various emerging applications, such as artificial muscles, soft actuators, and biomedical devices, owing to their unique shape recovery-induced contraction force. However, the factors influencing this force remain unclear. Herein, we designed a simple polymer blending system using a series of tetra-branched poly(ε-caprolactone)-based SMPs with long and short branch-chain lengths that demonstrate decreased crystallinity and increased crosslinking density gradients. The resultant polymer blends possessed mechanical properties manipulable across a wide range in accordance with the crystallinity gradient, such as stretchability (50.5–1419.5%) and toughness (0.62–130.4 MJ m^−3^), while maintaining excellent shape-memory properties. The experimental results show that crosslinking density affected the shape recovery force, which correlates to the SMPs’ energy storage capacity. Such a polymer blending system could provide new insights on how crystallinity and crosslinking density affect macroscopic thermal and mechanical properties as well as the shape recovery force of SMP networks, improving design capability for future applications.

## 1. Introduction

Shape-memory polymers (SMPs) are representative smart polymers and have been extensively studied in the past few decades owing to their unique capability of recovering their permanent original shapes from a temporarily fixed shape when actuated by various external stimuli, such as heat [1,2,3], light [4,5], lasers [6,7], and magnetic fields [8,9]. Immense efforts have been dedicated to exploring this unique shape-memory property and its induced shape-memory motions, such as bending, stretching, lifting, and grasping, in vast applications, including soft robotics, artificial muscles, actuators and biomedical devices [10,11,12,13,14,15]. While the appealing shape-memory motions explored so far have shown great potential, the feasibility of SMPs in practical applications suffers from various limitations, such as inadequate deformability, maneuverability, and shape recovery-induced contraction force.

Wang et al. [16] reported a light-actuated SMP composed of Poly(ε-caprolactone) (PCL)-based semicrystalline polymer network and polydopamine nanospheres as photothermal filler. The proposed artificial muscle-mimicking action was achieved by the shape-recovery-induced contraction motion, which was demonstrated by lifting a weight. Similarly, Li et al. [17] reported an SMP capable of mechanical bearing achieved by solvent-induced shape recovery force, which is applicable for soft robotics, sensors, etc. Kumar et al. [18] demonstrated an SMP-based smart compression device that utilizes the shape recovery-induced contraction force for the treatment of chronic venous disorder. In our previous study, we designed a novel shape-memory string capable of occluding the feeding vessels of fetal tumors in minimally invasive fetoscopic surgery contributed by the laser-actuatable shape-memory contraction [19]. The aforementioned shape recovery force is a pivotal function of SMPs, which are commonly utilized in wide range of applications. However, aside from demonstrations of such motion, such as weight-lifting [12] and muscle-mimicking contractions, the shape recovery force and its influential factors have not yet been systematically investigated.

PCL is a semicrystalline SMP with a melting temperature (T_m_) in the vicinity of 60 °C. Figure 1a shows a schematic illustration of the shape-memory mechanism for PCL-based SMPs. When heat above its T_m_ is applied to the semicrystalline SMP, accompanied by melting of the crystal regions, the molecular mobility of the chains increases. At this time, the SMP can be deformed by the application of external stress. Upon cooling below its crystallization temperature (T_c_), the recrystallization process can contribute to the weakening of the chain mobility, which enables the storage of entropic energy in the SMP in the form of elastic potential energy. The stored energy can then be released upon reheating, which reactivates the chain mobility and drives it back into its original shape, which is the highest entropic state [20,21].

Extensive research has shown that the semicrystalline feature of PCL contributes to a tunable mechanical property, which is greatly dependent on its crystallinity. Several attempts have been made to substantiate the crystallinity effect on mechanical properties. Miyasako et al. [22] reported a copolymer of *ε*-caprolactone (CL) and lactic acid, which endowed a drastic decrease in crystallinity due to the increased lactic acid ratio in the copolymers. The resultant copolymer exhibited weakened shape-memory characteristics. A similar copolymerization approach has been utilized by multiple recent studies, such as Xu et al. [23], who reported a copolymer of CL and n-butyl acrylate that demonstrated decreased crystallinity. A simple alternative approach would be blending PCL with other polymers with low crystallinity to tune the crystallinity of the polymer network. Chen et al. [24] blended PCL with poly(L-lactide), and the resultant polymer blends showed crystallinity-dependent mechanical properties as well. We have reported a polymer blending approach of tuning the crystallinity using a polymer blend of tetra-branched and linear PCL [25]. Concerning the immiscibility when blending with other types of polymers, this blending approach achieved successful tuning of the crystallinity and obtained the design capability of polymer blends with a T_m_ between 30.7 °C and 42.7 °C without the loss of shape-memory characteristics.

Herein, we designed and fabricated a series of shape-memory polymer blends (SMPBs) using semicrystalline PCL-based SMPs that can provide clear crystallinity and crosslinking density gradients (Figure 1b). Tetra-branched PCL (4bPCL) has been reported to possess less entanglement and provide better crosslinking efficiency compared to linear PCL. Therefore, a stable thermally crosslinked polymer blend network was obtained by simply blending two types of 4bPCL macromonomers with long (high crystallinity) and short (low crystallinity) chain lengths, thermally initiated by benzoyl peroxide (BPO) and the acryloyl end group on a PCL macromonomer. Low molecular weight 4b10PCL is theoretically amorphous after crosslinking, resulting from the short chain length between crosslinking points, which can contribute to a higher crosslinking density and usually results in difficulty forming crystal regions [26]. Contrarily, high molecular weight 4b100PCL is highly crystalline and possesses excellent mechanical properties, as reported in our previous study [19]. By simply blending different ratios of amorphous 4b10PCL with highly crystalline 4b100PCL, we could obtain a series of polymer blends that could help distinguish the influence of crystallinity and crosslinking density on macroscopic thermal, mechanical, and shape-memory properties such as elastic modulus, stretchability, T_m_, and shape recovery force, as shown in Figure 1c. With a better understanding of the influences of crystallinity and crosslinking density on the shape recovery force of semicrystalline SMPs, we can expect great convenience and precision in the future design of soft robotics, actuators, and biomedical devices.

## 2. Materials and Methods

Materials. All reagents were used as received unless otherwise specified. Pentaerythritol, acryloyl chloride, tin (II) 2-ethylhexanoate (Sn(Oct)_2_), CL, and triethylamine were purchased from Tokyo Chemical Industry Co., Ltd., Tokyo, Japan. Xylene, tetrahydrofuran (THF) (super-dehydrated), hexane, acetone, diethyl ether (super-dehydrated), methanol, and chloroform-d (CDCl_3_) with 0.05 *v*/*v*% tetramethylsilane (TMS) were purchased from Wako Pure Chemical Industries Ltd., Osaka, Japan. BPO was purchased from Sigma-Aldrich, St. Louis, MO, USA.

SMP synthesis. 4bPCL. Tetra-branched initiator pentaerythritol (168.93 mg, 1.25 mmol) was pre-dried in a 500 mL flask overnight prior to use. The monomer CL (52.84 mL, 0.5 mol) was added by a glass syringe under a nitrogen atmosphere with a catalytic level of Sn(Oct)_2_. The polymerization was carried out at 120 °C for 12 h under a nitrogen atmosphere. Afterward, the reacted solution was diluted with 300 mL of THF and reprecipitated with 1700 mL of hexane/diethyl ether mixture (1:1 *v*/*v*). Residual solvent was removed by filtration, and the precipitate was dried under reduced pressure overnight. The precipitate was repeatedly purified with the same mixture solvent three to five times, and purified 4b100PCL was collected (98.2% yield). The 4b10PCL was synthesized using pentaerythritol (1.69 g, 12.5 mmol) and CL (52.84 mL, 0.5 mol) with the same approach (97.9% yield). The synthesis scheme is shown in Appendix A (Appendix A).

4bPCL macromonomers. First, 100.0 g (2.18 mmol) of obtained 4b100PCL was completely dissolved in 500 mL of THF by ultrasonication. Excessive amounts of acryloyl chloride (1.862 mL, 20.6 mmol) and triethylamine (3.91 mL, 38.6 mmol) were added under ice bath in a shaded environment to react for 12 h. The reaction solution was reprecipitated with 1500 mL of methanol. The reprecipitation process was repeated three to five times, after which the 4b100PCL-macro was dried under reduced pressure and collected. The 4b10PCL-macro was synthesized in a similar manner with corresponding molar ratios.

Polymer characterization. The chemical structures, degree of polymerization of the 4bPCLs, and the end group introduction rate (I.R.) of 4bPCL-macros were determined by proton nuclear magnetic resonance spectroscopy (^1^H NMR) (JEOL, Tokyo, Japan) in CDCl_3_. The representative ^1^H NMR spectra of 4b10PCL and the 4b10PCL macromonomer are shown in Appendix A (Appendix A). The molecular weights were determined by gel permeation chromatography (GPC, JASCO International, Tokyo, Japan) using DMF as the elution solvent and polystyrenes as the calibration standard. The representative GPC trace of 4b1-PCL is shown in Appendix A (Appendix A). The results of molecular weights, polydispersity indexes (PDI), and I.R. of the 4b10PCL and 4b100PCL macromonomers are summarized in Appendix A (Appendix A).

SMPB film fabrication. SMPB films were fabricated using thermal initiator BPO and the previously synthesized 4bPCL-macros. The 4bPCL-macros (500.0 mg) were dissolved in xylene (40 wt.%), after which BPO (25.0 mg, 5 wt.%) was added under ultrasonication. The mixture was casted with a Teflon spacer (30 mm × 30 mm × 0.3 mm) sandwiched by two glass slides. The reaction was carried out at 80 °C for 6 h. Next, the crosslinked SMPB films were demolded and immersed in acetone for 2 days to remove the unreacted compounds. The obtained films were then washed with methanol for 2 h before being dried under reduced pressure overnight. Dried SMPB films were annealed at 60 °C in an oven for 10 min and collected after cooling down at room temperature [19].

Swelling ratio measurement. Crosslinked cylindrical samples were prepared under the aforementioned film fabrication conditions using 0.3-mm-diameter capillary tubes as molds corresponding to the 0.3-mm Teflon spacer for the SMPB films. The specimens were immersed in THF solutions for 48 h to reach the equilibrium swelling state. The diameters of the specimens were measured by laser scanning microscope (VK-X1000, Keyence Co., Itasca, IL, USA) at five different spots. The swelling ratio (Q) of each sample was estimated by its volume change according to the equation Q = (L_s_/L_0_)^3^, where L_s_ and L_0_ are the diameter of the specimen at the equilibrium swollen state and the initial diameter of the specimen, respectively. A minimum of three samples from each condition were tested in all tests to confirm reproducibility.

Thermal characterization. The thermal properties of SMP films were characterized using a differential scanning calorimetry (DSC) machine (7000X, Hitachi High-Tech Science, Tokyo, Japan). All samples were first equilibrated at 100 °C and cooled to −30 °C. The DSC curves were obtained in the second heating run at a rate of 5 °C min^−1^. The degree of crystallinity (χ_c_) was calculated according to the equation χ_c_ = △H/△H_m_, where △H is the enthalpy change of melting for each SMPB blend film and △H_m_ is the melting enthalpy for 100% crystalline PCL, which is considered to be 136 J g^−1^ according to the literature [27].

Mechanical characterization. The mechanical properties of the SMPB films were characterized by uniaxial tensile tests using a tensile tester (EZ-S 500N, Shimadzu, Kyoto, Japan) equipped with a heating chamber (M-600FN, TAITEC, Koshigaya, Japan). All experiments were carried out at an elongation rate of 5 mm min^−1^ at 20 °C or 60 °C. All specimens had 4.5-mm width, 30.0-mm length, and 0.3-mm thickness. For each sample, a minimum of three specimens were tested. The elastic modulus was calculated from the initial slope of the stress–strain curve. Toughness was calculated from the integrated area under the stress–strain curves until break. The integrated area under the stress–strain curves at each stretched strain at 60 °C was defined as the energy storage capacity at that deformation rate.

Shape-memory characterization. The 4b100PCL/4b10PCL polymer blend films were equilibrated at 70 °C, stretched to 800% strain (for 4b100PCL/4b10PCL polymer blends with 0–30 wt.% of 4b10PCL ratios within their deformation range), and cooled at 0 °C. Next, the external force was removed. The shape recovery was actuated by reheating at 70 °C. The shape fixity ratios (R_f_) of the 4b100PCL/4b10PCL SMPB films were calculated according to the equation R_f_ = 100% × ɛ/ɛ_load_, where ɛ represents the fixed strain after the removal of external force and ɛ_load_ represents the initial stretched strain. The shape recovery ratios (R_r_) of the 4b10/4b100 PCL blend films were calculated as R_r_ = 100% × (ɛ − ɛ_rec_)/ɛ, where ɛ_rec_ represents the strain after recovery [28]. The shape recovery-induced contraction force was measured using a strain-controlled Dynamic Mechanical Analyzer (ARES G2, TA Instruments, New Castle, DE, USA) with a designed program. The 4b100PCL/4b10PCL polymer blend films were equilibrated at X °C (each SMP’s T_m_ + 10 °C) at the speed of 10 °C min^−1^, held at equilibrium for 2 min, and stretched to different strains (50%, 100%, 200%, 300%, 500%, and 800% for each polymer blend within its deformation range) at a speed of 1 mm min^−1^. They were then cooled down to Y °C (each SMP’s T_c_ −20 °C) at the speed of 10 °C min^−1^, held at equilibrium for 5 min, and then reheated to X °C at maximum speed. The shape recovery-induced contraction force was recorded as the instantaneous force change above each sample’s T_m_. A minimum of three samples from each condition were tested in all tests to confirm reproducibility. 

## 3. Results and Discussions

### 3.1. Thermal Properties

The thermal properties of semicrystalline polymers have been reported to be greatly influenced by their crystallinity. For the substantiation of the crystallinity effect of the 4b100PCL/4b10PCL SMPBs, their thermal properties were evaluated by DSC. The endothermic melting peaks of 4b100PCL/4b10PCL SMPB films in Figure 2a demonstrated clear decreasing peak positions with the increase of the 4b10PCL ratios. The same decreasing positions were observed at the exothermic crystallization peaks in Figure 2b. The T_m_, T_c_, melting enthalpy (ΔH_m_), crystallization enthalpy (ΔH_c_) and calculated degree of crystallinity (χ_c_) are summarized in Appendix A (Appendix A). The T_m_ and T_c_ of the crosslinked 4b100PCL homopolymer were 57.3 °C and 27.7 °C, respectively, while the crosslinked 4b10PCL homopolymer is theoretically amorphous and showed a shallow melting peak at around 20.5 °C and a crystallization peak at around −19.1 °C. Figure 2c,d demonstrates that the increasing blending ratios of amorphous 4b10PCL can be attributed to a clear decreasing tendency of crystallinity from 39.41–10.66%. Consequently, T_m_ and T_c_ both showed excellent maneuverability with a large design capability (20.5–57.3 °C for T_m_ and −19.1 to 27.7 °C for T_c_), which proves the crystallinity-dominant effect on thermal properties of the polymer blending systems, which is consistent with previous literature [26,29]. Similar results were substantiated by the X-ray diffraction (XRD) patterns of the 4b100PCL/4b10PCL SMPBs (Appendix A, Appendix A). Two distinct diffraction peaks were observed for all blend polymer samples at 2θ = 21.6° and 2θ = 23.8°, which respectively correspond to the (110) and (200) planes of orthorhombic crystalline structures of PCL [30]. The decreasing peak areas under the XRD patterns of the 4b100PCL/4b10PCL SMPBs substantiated the decreasing crystallinity gradient of the 4b100PCL/4b10PCL SMPBs.

### 3.2. Mechanical Properties

A number of mechanical properties, including elastic modulus at room temperature and toughness, are crucial for the practical application of semicrystalline SMPs in a variety of fields. In these polymer blending systems, various mechanical properties were investigated to determine the influences of crystallinity and crosslinking density. Figure 3a provides a brief schematic illustration of the decreasing crystallinity gradient of the 4b100PCL/4b10PCL polymer blending systems. As substantiated in the previous section, the increase of the short chain amorphous 4b10PCL ratio could have led to the hindrance in forming the crystal region, resulting in the clear decreasing crystallinity gradient in the polymer blending systems. The correlations between this crystallinity change and the mechanical properties were investigated by uniaxial tensile tests. The elastic moduli, toughness, and stress–strain curves of the 4b100PCL/4b10PCL polymer blending systems in the uniaxial tensile test at room temperature are shown in Figure 3b,c. The related data are summarized in Appendix A (Appendix A). The results revealed a drastic decrease in elastic modulus as well as toughness at room temperature, which correspond to the decline in the degree of crystallinity. Compared to the elastic modulus of the 4b100PCL homopolymer (approximately 250.4 MPa), that of the 4b10PCL homopolymer was extremely low (2.6 MPa) due to its amorphous feature. This proved that crystallinity is the dominant factor in determining the stiffness and toughness of the semicrystalline SMPs. In addition, the decrease in crystallinity resulted in the decline in strain at the break of the polymer blending systems, as depicted in the typical stress–strain curves of the 4b100PCL/4b10PCL polymer blend films in Figure 3d. Moreover, the 4b100PCL/4b10PCL polymer blend films with 0–50 wt.% of 4b10PCL ratios all presented a typical stress–strain curve of thermoplastic polymer with a clear yielding point, necking feature, and a hardening process, which accord with the previous literature [30]. Furthermore, the 4b100PCL/4b10PCL polymer blend films with 60–90 wt.% of 4b10PCL ratios presented highly crosslinked rubber-like tensile features with relatively short strains at break, in accordance with the 4b10PCL homopolymer.

In addition to the mechanical properties at room temperature investigated above, which were proven to be mainly dominated by the crystallinity of the polymer blending systems, the mechanical properties above T_m_ were of great significance. These include the elastic modulus and stretchability at 60 °C, at which temperature the influence of crystallinity is expected to be omitted due to the elimination of crystal regions above all samples’ T_m_.

As briefly sketched in Figure 4a, the shorter-branched feature represents 4b10PCL, and the longer one represents 4b100PCL. The incorporation of short-chain 4b10PCL with crosslinkable end groups is thought to contribute to a highly crosslinked polymer network. Thus, with the increase of the incorporation amount of 4b10PCL, it is inferred that the 4b100PCL/4b10PCL polymer blending systems possesses an enhanced crosslinking density.

The swelling ratios of the 4b100PCL/4b10PCL SMPB films were measured in THF, a good solvent for PCL, for 48 h (until equilibrium) to confirm the hypothesized increase in the crosslinking density gradient of the aforementioned systems. As shown in Figure 4b, owing to the increase in crosslinking density, the swelling ratio showed a decreasing tendency with regards to the 4b10PCL ratios, which is in accordance with the previous hypothesis. For further confirmation of the increasing crosslinking density gradient of the blend polymer systems, the above T_m_ mechanical properties were investigated using uniaxial tensile tests at 60 °C. The elastic moduli and stretchability of the 4b100PCL/4b10PCL polymer blending systems are shown in Figure 4c, and related data are summarized in Appendix A (Appendix A). The typical stress-strain curves of the 4b100PCL/4b10PCL polymer blending systems are compared in Figure 4d, and the stress–strain curves of the 4b100PCL/4b10PCL polymer blend films with 50–100 wt.% of 4b10PCL ratios are shown in Figure 4e. A drastic decline in the stretchability of the blend polymers from 1419.5% for 4b100PCL to merely 50.5% for 4b10PCL was observed. When increasing the low molecular weight 4b10PCL ratios in the 4b100PCL/4b10PCL SMPBs, the chain length between crosslinking points drastically decreases, resulting in an elevated crosslinking density and an extensive decline in chain mobility which led to this almost 30-fold difference in stretchability. On the same grounds, a steady inclining tendency of the elastic modulus at 60 °C from approximately 0.58 MPa for 4b100PCL to around 2.05 MPa for 4b10PCL was found, which was attributed to the highly crosslinked polymer network. 

### 3.3. Shape-Memory Properties

The crystallinity and crosslinking density effects on macroscopic thermal and mechanical properties were probed in the previous sections. The ultimate purpose of understanding the correlations of the aforementioned parameters and their influences on the shape recovery force are discussed in this section. During the shape-memory cycle, after deformation upon heating, the recrystallization process can provide fixation of the temporary secondary shape due to the weakening of the chain mobility. In turn, the entropic energy that was endowed during the heating and deformation process can be stored inside the SMPs in the form of elastic potential energy. The amount of the energy stored inside, called the energy storage capacity, can be released upon the reactivation of chain mobility during the reheating, which could be the main driving force of the shape recovery force. Therefore, the energy storage capacity and shape recovery force of the 4b100PCL/4b10PCL polymer blending systems were investigated to obtain a better understanding of their correlations and dominant influential factors.

In our previous study, we reported that the increase in stretched strains could give rise to a stronger shape recovery force and better contraction effect [19]. In support of this conclusion, the shape recovery forces and energy storage capacities at different stretched strains of 4b100PCL are depicted in Figure 5a. The shape recovery force of the 4b100PCL homopolymer greatly enhanced from 0.061 N at 50% strain to 0.894 N at 800% strain. These results are in accordance with the previous conclusion that an almost linear enhancement in shape recovery force that correlates with the energy storage capacity in response to the increase of the stretched strains occurs [19,31]. Moreover, evidence suggests that crosslinking density is among the most important factors for tuning the shape-memory properties [28]. On this basis, due to the poor stretchability of 4b10PCL SMP film, the investigation of the influential effect was conducted by comparing the shape recovery forces and energy storage capacities of 4b100PCL/4b10PCL polymer blend films stretched at 50% strain. Figure 5b shows a clear inclining correlation of both shape recovery force and energy storage capacity with regards to the increasing 4b10PCL ratio, which corresponds to the crosslinking density gradient. On this basis, we concluded that shape recovery force is mainly dominated by the crosslinking density instead of the degree of crystallinity in the polymer blending system when stretched at the same strains. For further confirmation of this preliminary result, the shape recovery forces of 4b100PCL/4b10PCLSMPBs with 0–30 wt.% of 4b10PCL ratios at a small stretching strain of 50% to a large deformation of 800% were measured (Figure 5c). The summarized results also demonstrated a clear strain-dependent enhancement of the shape recovery force. Furthermore, the shape recovery forces were all gradually elevated with regards to the 4b10PCL ratios at every strain, which proved the dominance of the crosslinking density effect in such a property. Large deformations such as 800% have been commonly reported to greatly deteriorate shape-memory properties. Therefore, the shape fixity and shape recovery ratios of the 4b100PCL/4b10PCL polymer blends with 0–30 wt.% of 4b10PCL ratios stretched at 800% are shown in Figure 5d. The results showed that all samples possessed excellent shape fixities above 99.00%. A slight increase from 99.05% shape fixity for 4b100PCL to 99.73% for the 4b100PCL/4b10PCL_70/30 polymer blending system was observed. Contrarily, the shape recovery ratio decreased from 97.67% for 4b100PCL to 88.89% for the 4b100PCL/4b10PCL_70/30 polymer blending system. The changes in shape-memory properties were most likely due to the highly crosslinked polymer network at large deformations resulting in an alignment in the crystal region during recrystallization, which contributes to a relatively better fixation and energy storage and slightly weakened recovery due to the decrease in total crystallinity. However, the shape-memory properties were still excellent despite the large deformation. Therefore, a simple manipulation of SMPB system using minor amount of short-chain 4b10PCL can enhance the shape recovery force by approximately 166% (0.061 N for 4b100PCL homopolymer and 0.162 N for 4b100PCL/4b10PCL_70/30 SMPB at 50% stretched strains) without compromising much of shape-memory performance.

## 4. Conclusions

In this study, we designed a simple polymer blending system using a series of 4bPCL-based SMPs with long and short branch chain lengths that demonstrate decreased crystallinity and increased crosslinking density gradients. The influences of crystallinity and crosslinking density on the polymer blends were investigated, and the results confirmed the crystallinity-dominant effect on thermal properties and mechanical properties at room temperature, such as elastic modulus, stretchability, and toughness. The crosslinking density-dominant effect was shown in the mechanical properties at 60 °C at the shape recovery force of the SMPBs. With the understanding gained from this study, we could achieve a series of polymer blending systems with highly manipulable mechanical properties covering a wide range, such as stretchability (50.5–1419.5%) and toughness (0.62–130.4 MJ m^−3^), while maintaining excellent shape-memory properties even at a large deformation of 800%. Although the shape recovery force was proven to be dominated by the crosslinking density of the polymer network, the stretchability was also decreased in response to the decreased chain length between crosslinking points due to the elevated crosslinking density. Therefore, balanced manipulation between crystallinity and crosslinking density in the polymer network of SMPs is required for the design of an appropriate range for the shape recovery force. The elucidation of the factors influencing each property gained from this study is expected to contribute to better design of semicrystalline SMP networks, in turn providing great potential for future applications in soft robotics, actuators, and biomedical devices.

## Figures and Tables

**Figure 1 polymers-14-04740-f001:**
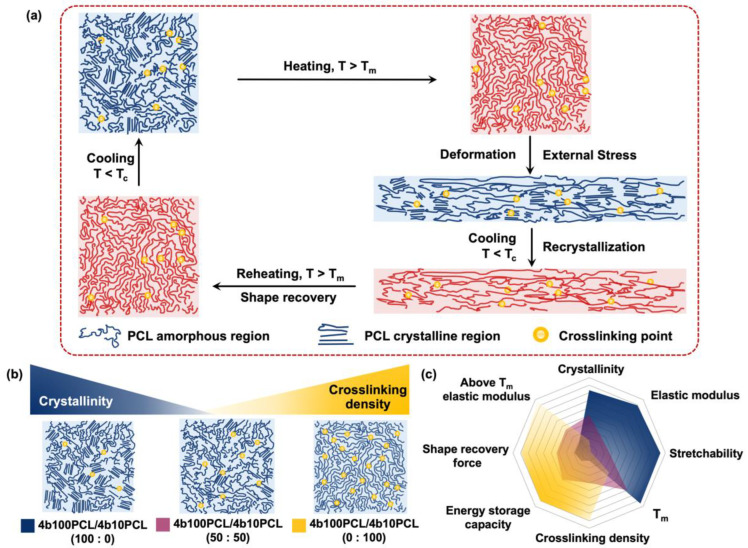
(**a**) Shape-memory mechanism for semicrystalline PCL-based SMPs. (**b**) Decreased crystallinity gradient and increased crosslinking density gradient achieved by the manipulation of amorphous 4b10PCL ratios in 4b100PCL/4b10PCL polymer blending systems. (**c**) Comparison of major physical properties and their expected correlations with crystallinity and crosslinking density in 4b100PCL/4b10PCL polymer blending systems. Different 4b100PCL/4b10PCL ratios are indicated by different colors.

**Figure 2 polymers-14-04740-f002:**
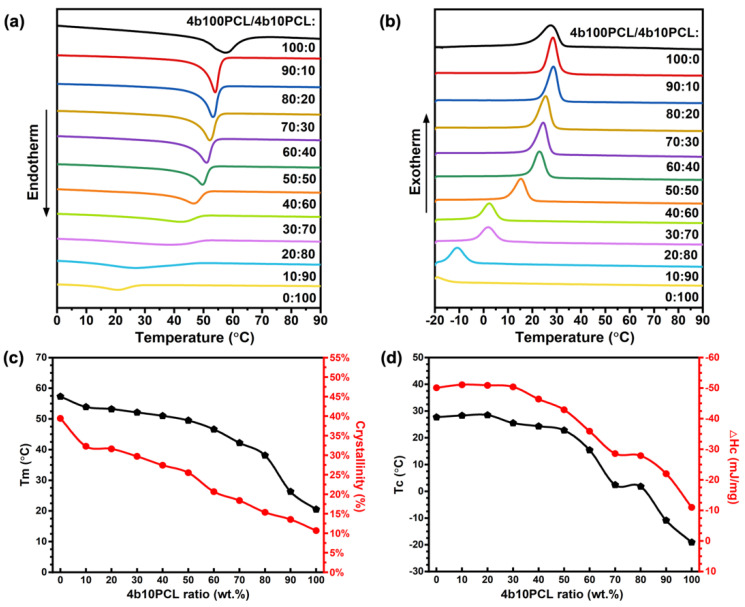
Thermal properties of 4b100PCL/4b10PCL polymer blends prepared with different 4b10PCL ratios. (**a**) DSC melting curves, (**b**) DSC crystallization curves, (**c**) melting temperature and crystallinity, and (**d**) crystallization temperature and ΔH_c_ of the 4b100PCL/4b10PCL polymer blends.

**Figure 3 polymers-14-04740-f003:**
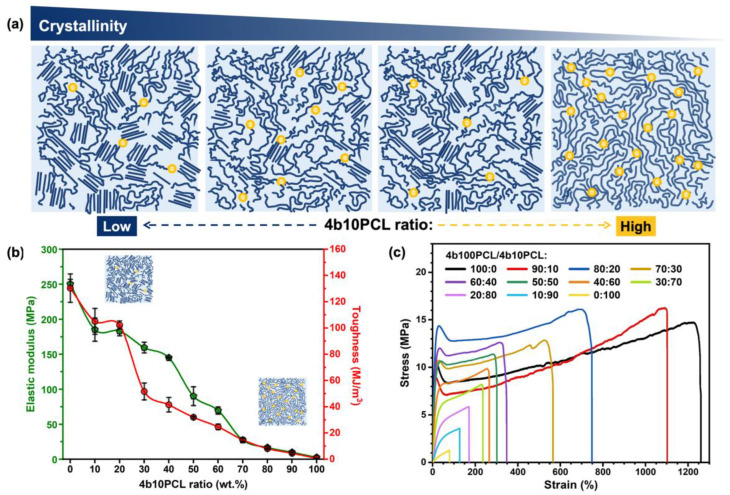
Crystallinity-dominant effect on the mechanical properties of 4b100PCL/4b10PCL polymer blends at 20 °C. (**a**) Schematic illustrations of the decreasing crystallinity of the 4b100PCL/4b10PCL polymer blends with the increase of the amorphous 4b10PCL ratio, (**b**) Elastic modulus and toughness, and (**c**) stress-strain curves of the 4b100PCL/4b10PCL polymer blends with gradient 4b10PCL ratios.

**Figure 4 polymers-14-04740-f004:**
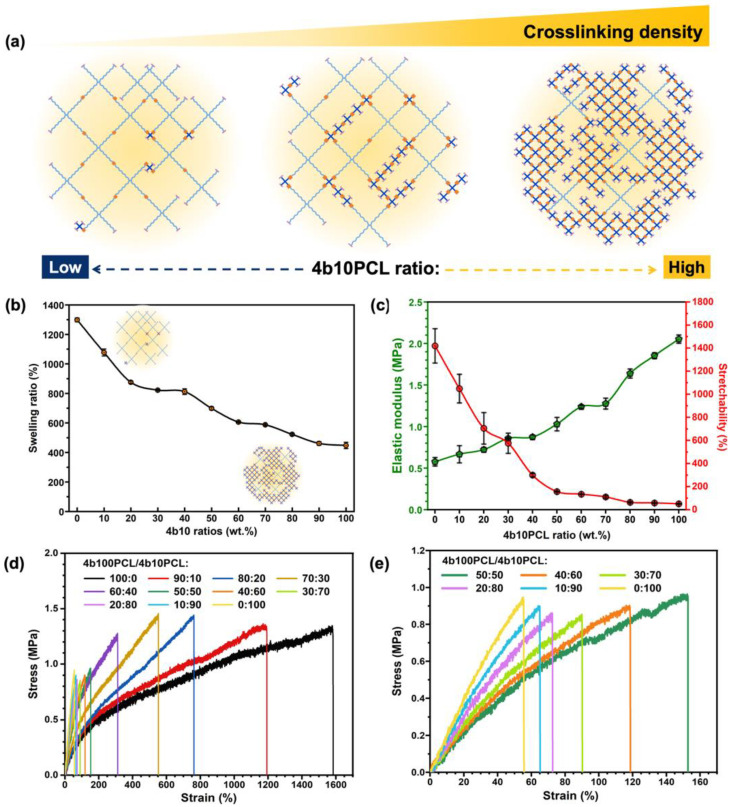
Crosslinking density-dominant effect on the mechanical properties of 4b100PCL/4b10PCL polymer blends at 60 °C. (**a**) Schematic illustrations of increasing crosslinking densities of the 4b100PCL/4b10PCL polymer blends with the increase of the short-chained 4b10PCL ratios, (**b**) swelling ratios measured at room temperature, (**c**) elastic modulus and stretchability, (**d**) stress–strain curves of the 4b100PCL/4b10PCL polymer blends with gradient 4b10PCL ratios, and (**e**) zoomed in stress–strain curves of the 4b100PCL/4b10PCL polymer blends with 0–50 wt.% of 4b10PCL ratios.

**Figure 5 polymers-14-04740-f005:**
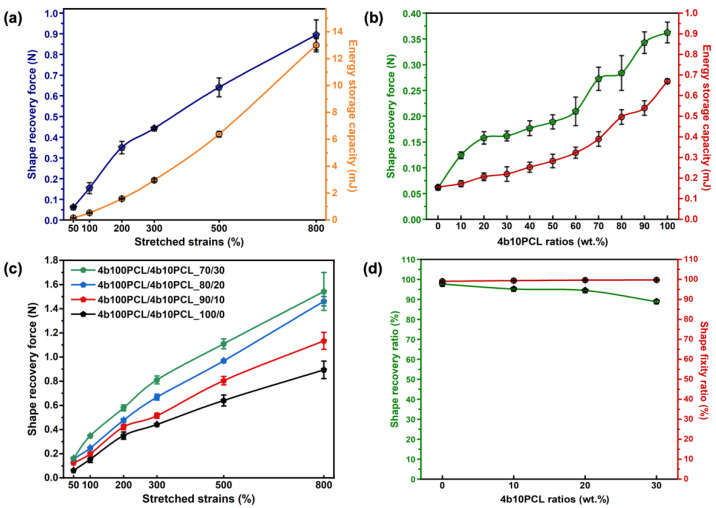
Shape-memory properties, shape recovery forces, and their correlations with the energy storage capacities of the 4b100PCL/4b10PCL polymer blends. Shape recovery force and energy storage capacities of (**a**) 4b100PCL at different stretched strains and (**b**) 4b100PCL/4b10PCL polymer blends with gradient 4b10PCL ratios at 50% stretched strain. (**c**) Shape recovery forces of 4b100PCL/4b10PCL polymer blends with 0–30 wt.% of 4b10PCL ratios at different stretched strains. (**d**) Shape recovery ratios and shape fixities of 4b100PCL/4b10PCL polymer blends with 0–30 wt.% of 4b10PCL ratios at a large deformation of 800%.

## Data Availability

Not applicable.

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
