# Peer review of "Influences of Crystallinity and Crosslinking Density on the Shape Recovery Force in Poly(ε-Caprolactone)-Based Shape-Memory Polymer Blends"

_polymers, 2022, doi:10.3390/polym14214740_

Round 1

Reviewer 1 Report

This manuscript reports the synthesis and characterization of tetra-branched poly(e-CL) in order to clarify the factors influencing restoring force and mechanical properties.  The work seems well executed, but, unfortunately the results are not novel and are completely expected.  In short, all the expected trends are observed.  There is a large amount of literature that reports the same trends for almost a decade.  Unless there is a new result that adds to the science or engineering of PCL based systems, I cannot support publication.  The authors may be interested in the full body of work by Mather's group (while at Syracuse) and Anthamatten's group (while Rochester) who reported essentially the same trends a while ago. 

Author Response

Dear Reviewer,

First of all, we would like to thank you for your constructive points which help to improve our manuscript. Please, find the attached revised version of our manuscript entitled, “Influences of Crystallinity and Crosslinking Density on the Shape Recovery Force in Poly(ε-caprolactone)-based Shape-memory Polymer Blends”. Additionally, all comments are taken into consideration and we hope our revision has improved the paper to the level of your satisfaction. The changes were highlighted in yellow color in the manuscript. Please also find below point-to-point responses to the criticisms in red color.

Thank you for your consideration. I look forward to hearing from you.

Sincerely,

Mitsuhiro Ebara

Reviewer 2 Report

The authors present a blend system formed by two star PCL polymers which are crosslinked through the thermal polymerization of acrylate groups at the end of the arms. The system allows for evaluating the effect of degree of crystallinity and crosslinking density, which are interdependent, on the mechanical properties and shape recovery forces. Thermal, mechanical and shape recovery characterization are sound and well described in general.

Nevertheless, the authors could consider the following observations:

+ L123: the amount of pentaerythritol is wrong. Please, correct it

+ SMPB film fabrication: has the mould a fixed cross section area? Please provide it

+ Swelling ratio measurement: are these samples polymerized inside 0.3 mm capillaries? or are 0.3 mm capillaries used as spacers instead of the 0.3 mm teflon spacer used in SMPB films? or are 0.3 mm capillaries used to punch 0.3 mm diameter cylinders out of the polymer film? Please provide a clearer explanation of the followed procedure

+ L209: please specify if 4b100PCL is crosslinked in this case as it is explicitly said for 4b10PCL in the same sentence.

+ Caption of Figure 4: “Crosslinking density dominant effect on the mechanical properties of 290 4b100PCL/4b10PCL polymer blends at 60ºC.” Was the swelling experiment in b performed at 60 ºC? If not, specify it.

+ Characterization by NMR:

- Chemical shifts should be provided and also the integration of the multiplet at 1,5-1,7 ppm corresponding to b+d protons

- Signal assignment seems wrong to me in both figures. Triplets at 2.3 ppm correspond to "a" protons , alpha to the C=O, and not to "c" protons. On the other hand the multiplet with the lowest chemical shift, at 1.4 ppm, corresponds to the "c" protons and not to "a". In Figure S1 the signal at 3.6 or 3.7 ppm does not correspond to the hydroxy group, but to the -CH2-OH (end group).

-Both in Figures S1 and S2, there are several options to set the integration. I would integrate "a" protons signal at 2.3 ppm -CH2-COO- as 20 (2 protons per r.u. and there are 10 r.u.).

- In this way you will obtain the integration of the signals corresponding to terminal groups: acrylate (already indicated in Figure S2), -CH2-OH (3.6 ppm, 2 H, in Figure S1) and, probably, -CH2-OOCCH=CH2 (small triplet at 4.1 ppm probably ca. 2 H, in Figure 2S).

- Parentheses of r.u. are missing in the chemical structure of Figure S2.

+ Characterization: GPC traces should be provided

+ ºC symbols are incorrect in several places (for example in L148)

+ Table S1: is "I.R.: End group introduction rate" the general way to refer to that parameter? Isn’t it the yield for the modification of the end groups? I think it would be appropriate to include (%) in any case.

Author Response

(The authors gave the same response as above.)

Reviewer 3 Report

The paper investigates the effects of crystallinity and crosslinking density on the thermal and mechanical properties as well as the constrained shape recovery performance of semi-crystalline polymers. The experimental results can well support the conclusions. I have the following comments for the current version of the paper:

1) Using the force to characterize the constrained shape recovery is inappropriate since different specimen size of the same materials may have different shape recovery force. The stress should be used instead.

2) It is also suggested to compare the maximum recovery stress with those in the literature, for example as shown https://doi.org/10.1016/j.ijplas.2019.102654; https://doi.org/10.1038/s41467-018-03094-2.             

3) The DSC tests can reveal the information on the melting transition as well as the glass transition behaviors. Can the authors provide any information regarding the glass transition of the polymers.  

4) The final recovery force/stress can actually be well estimated as the force/stress deformed to the same strain level at the high temperature. In general, the force/stress scales with the modulus at the high temperature and the applied strain. The authors are suggested to confirm whether the results are consistent with this information.

Author Response

(The authors gave the same response as above.)
